# Birth Outcomes in the Hispanic Population in the United States: Trends, Variation, and Determinants (2011–2021)

**DOI:** 10.3390/ijerph22091325

**Published:** 2025-08-26

**Authors:** Yanchao Yang, Sota Fujii, Thinh Nguyen

**Affiliations:** School of Business and Leadership, DePauw University, Greencastle, IN 46135, USA; sotafujii_2027@depauw.edu (S.F.); thinhnguyen_2026@depauw.edu (T.N.)

**Keywords:** Hispanic population, birth outcomes, C-section delivery, low birth weight, prematurity, maternal health disparities, classification tree, logistic regression

## Abstract

Infants born to mothers who self-identify as Hispanic account for a substantial and growing share of births in the United States, yet limited research has examined disparities in birth outcomes across Hispanic origin subgroups. This study aims to document trends and identify important factors associated with Cesarean section (C-section), low birthweight, and prematurity within the Hispanic population. We use data from the National Vital Statistics System (2011–2021), covering nearly all U.S. births. We compare outcomes across Hispanic, non-Hispanic White, and non-Hispanic Black mothers and further disaggregate by Hispanic origin (Mexican, Puerto Rican, Cuban, Central/South American, and Other/Unknown). We use logistic regression and classification tree models to assess associations between maternal, infant, and clinical factors and birth outcomes. We find that Hispanic mothers have birth outcomes similar to non-Hispanic Whites and better than non-Hispanic Blacks. However, prematurity rates among Hispanics have slightly increased over time. Mexican mothers exhibit the most favorable outcomes, while Cuban mothers show higher rates of C-section, and Puerto Rican mothers show higher rates of low birthweight and prematurity. Logistic regression results highlight multiple births, breech presentation, and hypertensive conditions as important factors associated with adverse birth outcomes. Our biomedical approach emphasizes physiological and clinical risk factors such as multiple births, breech presentation, hypertensive conditions, and obesity. In parallel, our biosocial analysis incorporates demographic, socioeconomic, and behavioral variables to contextualize how social determinants interact with biology to influence outcomes. Complementing these findings, our classification tree analysis identifies inadequate gestational weight gain (less than 15 pounds) as a prominent risk factor for both low birthweight and prematurity. Additionally, obesity emerges as a significant factor linked to an increased likelihood of C-section. While birth outcomes among Hispanic mothers are generally favorable, subgroup differences and emerging disparities highlight the need for disaggregated research and culturally tailored public health interventions.

## 1. Introduction

Infants born to mothers who self-identify as Hispanic accounted for 24% of all live births in the United States in 2021 (authors’ calculations), with this proportion showing a gradual upward trend [1]. Given the size and growth of this population, it is important to examine their birth outcomes in greater detail. Existing research has documented the “Hispanic paradox,” whereby Hispanic mothers, despite facing socioeconomic disadvantages—including lower educational attainment, limited access to medical care, and structural discrimination—often experience birth outcomes comparable to those of non-Hispanic White mothers [2,3,4,5,6,7]. However, concerns remain, including high rates of teen births, prenatal depression [8], and tobacco use during pregnancy among Hispanic women [9]. These patterns highlight the need for continued investigation into maternal and infant health within this population.

Research on birth outcomes among the Hispanic population often uses Cesarean section (C-section), low birthweight, and prematurity as key indicators [7,10,11,12]. A C-section is defined as the surgical birth of a fetus via incisions made in both the abdominal wall and the uterine wall [13]. Low birthweight is defined as a birth weight below 2500 g, and prematurity is defined as a gestational age of less than 37 weeks [14]. In current studies, these outcomes are frequently compared across racial and ethnic groups, particularly with non-Hispanic White and non-Hispanic Black mothers. Findings across studies are generally consistent: Hispanic and non-Hispanic White mothers tend to have similar rates of low birthweight—approximately 7.5% and 7%, respectively—and comparable rates of prematurity, around 9.6% for Hispanics and 9% for non-Hispanic Whites. In contrast, non-Hispanic Black mothers experience significantly higher rates of both low birthweight and prematurity (both approximately 14%) [14,15]. A similar pattern is observed in C-section rates, with both Hispanic and non-Hispanic White mothers at about 31%, compared to 36% among non-Hispanic Black mothers [14,16]. Studies examining birth outcomes across different countries of origin within the Hispanic population often focus on key subgroups such as Mexican, Puerto Rican, Cuban, Central or South American, and Caribbean origins [15,16,17]. These studies consistently document significant variation in birth outcomes across Hispanic origin groups. For example, Mexican mothers tend to have the lowest rates of prematurity and low birthweight compared to other Hispanic subgroups [5,15,17]. However, to the best of our knowledge, there is currently a lack of research examining variation in C-section rates across different Hispanic origin groups.

The factors associated with birth outcomes among the Hispanic population are multifaceted. They first include demographic and socioeconomic factors. Hispanic women experiencing compounded disadvantage, such as low income, limited education, and residing along the U.S.–Mexico border, had significantly higher odds of undergoing a Cesarean delivery compared to those without such disadvantage [18,19]. Another study showed that urban poverty was negatively associated with birthweight among Puerto Rican mothers but not among other Hispanic subgroups [5]. Additionally, lower individual educational attainment and lower county-level income were linked to increased odds of low-risk Cesarean deliveries [20]. Medicaid coverage had minimal impact on prematurity among Hispanic women [3]. Immigration status has emerged as a relevant factor: Latin American immigrant women demonstrated a higher risk of emergency C-sections compared to their non-immigrant counterparts [21]. Second, maternal health conditions and behavioral factors also play a critical role in shaping birth outcomes. Several studies have found that high levels of maternal depressive symptoms are common among Hispanic women enrolled in Medicaid [22]. Depressed women reported experiencing more negative life events compared to their non-depressed counterparts [8]. Women who experience depression during pregnancy are at increased risk of prematurity and low birthweight [23,24]. Hispanic women are also more likely to experience delayed initiation of prenatal care [25] and have higher rates of tobacco use during pregnancy [9].

These indicators—C-section, low birthweight, and prematurity—are essential public health metrics due to their strong associations with maternal and infant health outcomes [26,27,28,29]. A C-section is a critical obstetric intervention that can significantly reduce maternal and neonatal morbidity and mortality when medically necessary [30]. However, its increasing use in the United States has raised concerns about overuse [31,32]. Unnecessary C-sections can lead to higher risks of maternal mental health issues [33,34], complications in future pregnancies [35,36], and increased financial burdens for families [37,38]. Low birthweight serves as a key indicator of infant health. It is strongly associated with poor neonatal outcomes, a greater likelihood of NICU admission, and long-term developmental challenges [29,39]. Prematurity remains one of the leading causes of infant mortality and is linked to higher risks of both short- and long-term complications [40]. Given their clinical and policy relevance, we focus on these three outcomes to better understand maternal and infant health within the Hispanic population.

In this study, we build upon previous research by documenting the rates of C-section, low birthweight, and prematurity using data from the Natality Birth Files within the National Vital Statistics System (NVSS) from 2011 to 2021 [41], which captures nearly all births occurring in the United States. We compare birth outcomes among Hispanic mothers with those of non-Hispanic Black and non-Hispanic White. Additionally, we examine trends across Hispanic subgroups, including Mexican, Puerto Rican, Cuban, Central or South American, and Other when specific origin is not reported. To investigate the factors associated with these birth outcomes among Hispanic mothers, we further utilize the restricted-use version of the natality data. This allows us to incorporate a wide set of control variables categorized into four domains: demographic and socioeconomic characteristics, infant and birth-related factors, maternal behavioral factors, and maternal health conditions. We employ logistic regression analysis to examine the associations between various control variables and birth outcomes. In addition, we use classification tree analysis to explore the hierarchical structure and relative importance of these factors in predicting birth outcomes. Our aim is to provide a systematic and interpretable analysis of how these factors are associated with key birth outcomes within the Hispanic population.

Our study yields several key findings. First, the trends in C-section, low birthweight, and prematurity rates generally align with previous research. Hispanic mothers exhibit similar rates of these outcomes compared to non-Hispanic White mothers, while non-Hispanic Black mothers experience substantially higher rates across all three indicators. However, we observe a growing disparity in prematurity rates between Hispanic and non-Hispanic White mothers over the study period. Additionally, while the C-section rate among Hispanic mothers was initially lower than that of non-Hispanic Whites in 2011, it surpassed the non-Hispanic White rate by 2013.

Second, when disaggregating the data by Hispanic origin, we find considerable variation in birth outcomes. Cuban mothers consistently have the highest C-section rates, although this rate declined from 47% in 2011 to 44% in 2021. In contrast, Mexican mothers generally have the most favorable outcomes, with a C-section rate of approximately 31%, a low birthweight rate of 6.5%, and a prematurity rate around 11%. Starting in 2015, both low birthweight and prematurity rates show a slight upward trend across all Hispanic origin groups.

Third, our regression analyses reveal several consistent and subgroup-specific associations across Hispanic-origin groups. For Cesarean delivery, fetal presentation and prior C-section history show the strongest positive associations, while multiple births are negatively associated. Other notable risk factors include eclampsia, infertility treatment, fetal macrosomia, hypertensive disorders, and maternal obesity. Low birth weight is most strongly associated with multiple births, eclampsia, gestational and pre-pregnancy hypertension, and fetal presentation. Prematurity shares a similar risk profile with low birthweight but generally yields more modest effect sizes. Multiple births, hypertensive disorders, eclampsia, and diabetes are key contributors.

Our study contributes to the literature by offering a comprehensive and updated analysis of birth outcomes among Hispanic mothers using nationally representative data from 2011 to 2021. While prior research has often treated Hispanic mothers as a monolithic group, our findings emphasize the importance of disaggregating by Hispanic origin to uncover meaningful differences in outcomes and risk profiles. By incorporating a broad set of control variables and employing both logistic regression and classification tree models, we not only quantify the relative influence of medical and socioeconomic predictors but also offer interpretable decision pathways that help clarify the conditions most associated with adverse outcomes. This dual perspective allows us to frame our findings through both a biomedical lens—focusing on physiological and clinical risk factors—and a biosocial lens, which considers the ways in which demographic, socioeconomic, and behavioral contexts shape health outcomes. The biomedical approach highlights modifiable medical conditions and pregnancy-related complications that can be addressed through clinical interventions, whereas the biosocial perspective underscores structural and environmental determinants that require public health and policy solutions. By integrating these perspectives, our analysis bridges clinical medicine and population health, offering a more holistic understanding of disparities in maternal and infant outcomes. These insights can inform targeted public health interventions, clinical risk assessments, and culturally tailored maternal care strategies that address the unique needs of subpopulations within the Hispanic community.

## 2. Materials and Methods

### 2.1. Data Source and Sample Construction

The data used for this analysis comes from the Natality Birth Data within the National Vital Statistics System (NVSS), maintained by the National Center for Health Statistics (NCHS) [41]. This micro-level dataset is derived from information recorded on birth certificates filed with vital statistics offices in each U.S. state and the District of Columbia. The dataset captures nearly all births that occur in the United States and includes a rich array of demographic, socioeconomic, maternal health, and birth-related information. Specifically, it provides mothers’ demographic and socioeconomic status such as age, race, Hispanic origin, educational attainment, and marital status, as well as health-related characteristics including pre-pregnancy BMI, weight gain during pregnancy, timing of prenatal care initiation, number of prenatal visits, and diagnoses of gestational hypertension, diabetes, and infections (such as gonorrhea, or hepatitis B/C). The dataset also includes infant and birth characteristics, such as birthweight, gestational age, plurality, birth order, breech presentation, and use of C-section. We also obtained access to the confidential geographic information on the state of birth by applying through NCHS.

We constructed our sample as follows. We use data from 2011 to 2021 to analyze changes in C-section rate, low birthweight rate, and prematurity rate over time. We limit our sample to mothers who self-reported as Hispanics. To examine health outcomes across different Hispanic subgroups, we use a variable that categorizes mothers’ Hispanic origin into the following groups: Mexican, Puerto Rican, Cuban, Central or South American, and Other if not specified the exact Hispanic origin. Detailed information on race and ethnicity can be found in Appendix A. For the modeling analysis, we further exclude observations with missing values. The proportion of observations excluded due to missing data is reported in Table A1. Table 1 presents the summary statistics for the main outcome and control variables for the Hispanic population, with a further breakdown by Hispanic subgroups. For outcome trend analysis, we include non-Hispanic White and non-Hispanic Black mothers as comparison groups. We retain all observations for which information on C-section, low birthweight, gestational age, and race/ethnicity is available.

### 2.2. Measures

**Outcome variables:** We focus on three primary outcome variables. (1) C-section indicator, a binary variable indicating whether the delivery was performed via C-section; (2) Low birthweight indicator, defined as a binary variable equal to 1 if the infant’s birthweight is less than 2500 g; (3) Prematurity, defined as a gestational age of less than 37 weeks.

**Control Variables:** The study includes three broad categories of control variables, allowing us to examine how these factors are associated with health outcomes. The first group captures maternal demographic and socioeconomic characteristics, including the mother’s age, a binary indicator for having a bachelor’s degree or higher, marital status, and whether Medicaid was used as the primary payer for delivery. The second group focuses on infant birth characteristics and includes indicators for multiple births, whether the infant is the mother’s first birth, breech presentation, and fetal macrosomia, defined as a birthweight over 4000 g. The third group reflects maternal health characteristics, including an indicator for obesity based on pre-pregnancy body mass index (BMI), whether prenatal care was initiated during the first trimester, smoking status before and during pregnancy, diagnoses of gestational diabetes and hypertension, gonorrhea, Hepatitis B and Hepatitis C during pregnancy, and whether the mother received infertility treatment. Each of these variables is coded as 1 for “Yes” and 0 otherwise.

### 2.3. Data Analysis Strategy

#### 2.3.1. Documenting Trends in C-Section, Low Birthweight, and Prematurity Rates

To examine overall patterns, we begin by plotting trends in C-section rates, low birthweight rates, and prematurity rates for Hispanic, non-Hispanic White, and non-Hispanic Black mothers from 2011 to 2021. We then further disaggregate the data to show trends by specific Hispanic origin groups. This part of the data analysis was conducted using Python 3.10 on Google Colab.

#### 2.3.2. Logistic Regression

To examine the relationship between health outcomes and maternal demographic and socioeconomic characteristics, infant birth characteristics, and maternal health conditions, we employ logistic regression models, as all three outcome variables are binary [42]. The analysis is conducted using STATA 18. We include state fixed effects, year fixed effects, and day-of-week indicators for the birth date to account for unobserved geographic and temporal factors that may influence outcomes. We report the estimated odds ratios along with robust standard errors. Statistical significance is indicated at the 0.01, 0.05, and 0.10 levels based on *p*-values.

#### 2.3.3. Classification Tree

To provide an interpretable understanding of how control variables are associated with the three maternal and infant health outcomes—C-section, low birthweight, and prematurity—we conducted a classification tree analysis. This approach complements our regression models by offering a visualization of how different factors contribute to the likelihood of each outcome variable. A classification tree is a supervised machine learning method that predicts binary or categorical outcomes by recursively partitioning the data based on input variables [43]. In our analysis, the tree algorithm uses the Gini impurity as the criterion to evaluate each potential split. The Gini impurity measures the impurity of a node, with lower values indicating that the node contains a more homogeneous set of outcome categories. By minimizing the Gini coefficient at each split, the classification tree identifies the most informative variables and decision paths that differentiate between cases [43]—for example, whether a mother is more likely to have a C-section or not—based on demographic, health, and birth characteristics. This method allows us to highlight key variable interactions and thresholds in a way that is easily interpretable and practically meaningful for healthcare researchers and policymakers.

For each birth outcome variable, we constructed four separate classification tree models: one based on maternal demographic and socioeconomic characteristics, one on infant and birth characteristics, one on maternal health behaviors and physical health indicators, and one on maternal medical conditions and infections. Each tree was limited to three layers and eight terminal nodes to ensure interpretability. Our primary objective was to explore the hierarchical structure of associated factors rather than optimize predictive performance. These models help identify the most salient variables and their sequential influence on birth outcomes.

## 3. Results

### 3.1. Trends in C-Section, Low Birthweight, and Prematurity

As shown in Panel A of Figure 1, the C-section rate among Hispanic mothers remained relatively stable between 2011 and 2021, averaging around 32%. Compared to other racial groups, the rate for Hispanics is similar to that of non-Hispanic White mothers. However, while the C-section rate for non-Hispanic White mothers has shown a slight downward trend over time [44], the rate for Hispanics has remained flat. In contrast, non-Hispanic Black mothers consistently exhibited the highest C-section rates throughout the period. Panel B shows that the low birthweight rate among Hispanic mothers remained relatively stable over the study period, averaging around 7%, and closely aligned with the rate observed for non-Hispanic White mothers. Panel C illustrates that the prematurity rate for Hispanic mothers was approximately 11.8% in 2011 and increased slightly over time, reaching 12.3% by 2021. By the end of the period, this rate was about 2 percentage points higher than that of non-Hispanic White mothers.

Although the overall trends for the Hispanic group appear relatively stable, Figure 2 reveals substantial variation in health outcomes across different Hispanic origin subgroups. As shown in Panel A of Figure 2, Cuban mothers consistently have the highest C-section rates among all Hispanic origin groups, averaging around 47%, though this rate shows a slight decline over time. In contrast, Mexican mothers have the lowest C-section rates, which remain stable throughout the study period. Panels B and C indicate that Puerto Rican mothers exhibit the highest rates of low birthweight and prematurity, while Mexican mothers have the lowest rates for both outcomes. However, the rates for Mexican mothers show a slight upward trend over the study period.

### 3.2. Logistic Regression Results

Logistic regression estimates for C-section (N = 9,321,637), low birthweight (N = 9,324,315), and prematurity (N = 9,324,315) among Hispanic births observed between 2011 and 2021 are presented in Table 2, Table 3, and Table 4, respectively.

The C-section models (Table 2) reveal that fetal presentation (logOR = 3.50–3.80) and history of prior C-section (logOR = 3.22–3.61) represent the strongest predictors across all Hispanic-origin subgroups. Plurality demonstrates a negative association with Cesarean delivery (logOR = −1.89 to −2.25). Several risk factors show positive associations with increased odds of Cesarean delivery, including eclampsia (logOR = 0.79–1.04), infertility treatment (logOR = 0.74–0.85), fetal macrosomia (logOR = 0.53–0.68), gestational hypertension (logOR = 0.52–0.68), pre-pregnancy hypertension (logOR = 0.49–0.60), and pre-pregnancy obesity (logOR = 0.31–0.49). Subgroup analysis reveals distinctive patterns among Cuban mothers, consistent with the unique C-section rates observed in the trend analysis (Panel A of Figure 2). While maternal smoking (logOR = 0.06–0.21, excluding Cubans) and higher maternal education (logOR = 0.13–0.36, excluding Cubans) are associated with increased odds of Cesarean delivery across all other Hispanic subgroups, these variables demonstrate slightly protective effects among Cubans (logOR = −0.07 and −0.05, respectively). These findings align with the descriptive results showing that Cuban mothers exhibit relatively lower smoking rates and substantially higher rates of college education (Table 1). Medicaid coverage shows a marginal negative association with cesarean delivery across most subgroups (logOR = −0.03 to −0.16, excluding Cubans), while no significant association is observed among Cubans (logOR = 0.00). Given that Cubans demonstrate the lowest rate of Medicaid coverage (Table 1), this finding supports the hypothesis that Medicaid coverage may be positively correlated with the risk of C-section.

In the low birthweight models (Table 3), plurality emerges as the most influential predictor (logOR = −3.14 to −3.38), followed by eclampsia (logOR = 1.50–2.03), gestational hypertension (logOR = 1.19–1.53), pre-pregnancy hypertension (logOR = 1.07–1.44), and fetal presentation (logOR = 1.00–1.18). Additional significant predictors include maternal smoking (logOR = 0.77–1.02, excluding Puerto Ricans), infertility treatment (logOR = 0.30–0.36, excluding Cubans), hepatitis C (logOR = 0.20–0.44, statistically not significant among Cubans), and pre-pregnancy diabetes (logOR = 0.15–0.26). Pre-pregnancy obesity demonstrates a protective association with low birthweight (logOR = −0.34 to −0.55). Consistent with the trend analysis showing that Puerto Rican births exhibited the highest low birthweight rates throughout 2011 to 2021 (Panel B of Figure 2), this subgroup displays distinct risk factor patterns. Notably, the impact of smoking during pregnancy is attenuated among Puerto Ricans (logOR = 0.60), though it remains at a significantly high level. Among Cuban mothers, gonorrhea shows a stronger association with low birthweight (logOR = 0.58) compared to other subgroups (logOR = 0.10–0.17, excluding Cubans, statistically not significant among Puerto Ricans), while the effect of infertility treatment is comparatively diminished (logOR = 0.16). These patterns correspond to the descriptive findings of lower gonorrhea rates and higher infertility treatment utilization among Cubans (Table 1).

The prematurity models (Table 4) demonstrate patterns largely consistent with those observed for low birthweight, despite the trends of prematurity indicating higher and more volatile rates over time (Panel C of Figure 2). However, the estimated odds ratios are generally more modest in magnitude compared to the low birthweight models. Plurality remains the most influential predictor (logOR = −2.33 to −2.61), followed by eclampsia (logOR = 1.27–1.36), gestational hypertension (logOR = 0.85–1.05), pre-pregnancy hypertension (logOR = 0.82–0.94, excluding Cubans), and fetal presentation (logOR = 0.83–0.98, excluding Cubans). Additional factors significantly associated with increased odds of prematurity include pre-pregnancy diabetes (logOR = 0.54–0.80), maternal smoking (logOR = 0.44–0.58, excluding Puerto Ricans), infertility treatment (logOR = 0.31–0.37), and hepatitis C (logOR = 0.23–0.39). The patterns of prematurity among Cubans differed substantially from other Hispanic subgroups, exhibiting greater instability (Panel C of Figure 2). Among Cubans, pre-pregnancy hypertension (logOR = 0.71), breech presentation (logOR = 0.65), and infertility treatment (logOR = 0.19) demonstrate relatively attenuated associations compared to other Hispanic subgroups. Furthermore, while maternal education and timing of prenatal care initiation serve as significant protective factors against prematurity across all Hispanic subgroups (logOR = −0.16 to −0.18 and logOR = −0.14 to −0.20, respectively), these associations are substantially weaker among Cubans (logOR = −0.04 and logOR = −0.02, respectively). Puerto Ricans exhibit moderately reduced odds of prematurity associated with smoking (logOR = 0.31), consistent with their protective trend in prematurity outcomes relative to other Hispanic subgroups (Panel C of Figure 2) and despite having the highest smoking prevalence (Table 1).

### 3.3. Classification Tree Results

Figure 3, Figure 4, and Figure 5 present the classification tree results for Cesarean delivery, low birthweight, and prematurity, respectively, among Hispanic births.

Across all models predicting C-section delivery, distinct sets of predictors emerge depending on the domain of focus. The model using maternal demographic and socioeconomic characteristics as control variables (Figure 3a) shows that maternal age, educational attainment, and Medicaid usage as the most salient determinants, with older, non-college-educated, and publicly insured women exhibiting the highest probabilities of Cesarean delivery. The infant and birth characteristics model (Figure 3b) demonstrates stronger stratification, with previous C-section history, breech presentation, and singleton status driving the most pronounced increases in cesarean delivery rates. In the model using maternal health behavior and physical health as predictors (Figure 3c), obesity and excessive gestational weight gain (greater than 41 pounds) emerge as key contributors. The model using maternal medical conditions and infections information as controls (Figure 3d) shows gestational hypertension, both pregestational and gestational diabetes, and infectious conditions such as gonorrhea as important predictors. While each model independently highlights domain-specific risk factors, their combined interpretation reinforces the multifactorial nature of C-section risk, in which socio-structural vulnerabilities and clinical complications intersect.

As shown in Figure 4, the socioeconomic model (Figure 4a) identifies marital status and maternal age as primary predictors, with the highest probabilities of low birthweight observed among unmarried mothers and those under the age of 18. In contrast, the infant and birth characteristics model (Figure 4b) highlights different risk pathways, with low birthweight disproportionately concentrated among male infants, non-singleton births, and deliveries to mothers with no history of Cesarean section. The maternal behavioral and physical health model (Figure 4c) demonstrates strong predictive power for low gestational weight gain (less than 15 pounds) and obesity, while infertility treatment further stratified risk within weight gain categories. The maternal medical conditions and infections model (Figure 4d) emphasizes the role of hypertensive disorders, with gestational and chronic hypertension—particularly in combination with eclampsia—substantially increasing the likelihood of low birthweight.

In Figure 5, the socioeconomic model (Figure 5a) identifies marital status and maternal age as the most salient predictors, with unmarried and adolescent mothers demonstrating the highest probabilities of preterm birth. The birth characteristics model (Figure 5b) prioritizes singleton status, history of previous C-section, and breech presentation, suggesting that specific obstetric histories and fetal positioning contribute meaningfully to elevated prematurity risk. In the behavioral and physical health model (Figure 5c), inadequate gestational weight gain (less than 15 pounds), obesity, and absence of early prenatal care emerged as key contributors. Infertility treatment appears repeatedly across decision paths, indicating its nuanced role in moderating risk within different maternal contexts. The maternal medical conditions and infection complications model (Figure 5d) identified pre-pregnancy and gestational hypertension and eclampsia as high-risk pathways, with certain combinations of these conditions associated with sharply increased probabilities of premature birth.

## 4. Discussion

This study provides a comprehensive analysis of the determinants of Cesarean delivery, low birthweight, and prematurity among Hispanic births in the United States from 2011 to 2021. Our findings from both logistic regression models and classification tree analyses offer new insights into how clinical, behavioral, and sociodemographic factors intersect to influence these key birth outcomes within a heterogeneous Hispanic population.

Logistic regression results indicate that clinical risk factors are the strongest predictors across all outcomes. For C-section, fetal malpresentation and a history of prior Cesarean delivery were the most influential predictors, with particularly high log odds ratios across all Hispanic subgroups. Plurality was negatively associated with C-section, reflecting the lower likelihood of surgical delivery in non-singleton births under certain clinical conditions. Other consistent predictors include hypertensive disorders, pre-pregnancy obesity, and fertility treatments. Notably, subgroup analysis revealed distinct patterns among Cuban mothers, for whom maternal smoking and higher education showed slightly protective associations—deviating from trends observed in other Hispanic subgroups. These variations align with the descriptive statistics showing that Cubans have lower smoking rates and higher educational attainment.

For low birthweight, plurality, eclampsia, and hypertensive disorders were the strongest risk factors. Maternal smoking and hepatitis C also contributed significantly. Puerto Rican mothers—who consistently exhibited the highest low birthweight rates—showed attenuated effects of smoking, suggesting potential behavioral or biological differences. In contrast, among Cubans, the effect of infertility treatment was diminished, while gonorrhea was a stronger predictor of low birthweight than in other groups. Prematurity models revealed patterns largely consistent with low birthweight, though effect sizes were generally smaller. Plurality remained the dominant predictor, followed by eclampsia and hypertensive conditions. As with low birthweight, the predictive role of smoking and infertility treatment varied by subgroup. The Cuban subgroup again stood out with weaker associations between prematurity and key risk factors such as pre-pregnancy hypertension and delayed prenatal care.

Our classification tree models enrich these findings by illustrating how risk factors interact hierarchically. For C-sections, models showed that previous Cesarean, breech presentation [44], obesity, and gestational weight gain were dominant decision points, while the role of Medicaid usage, education, and maternal age underscored the socioeconomic dimensions of surgical delivery. For low birthweight and prematurity, classification trees highlighted inadequate gestational weight gain, non-singleton births, and hypertensive disorders as critical determinants. Marital status and adolescent pregnancy also emerged as strong socioeconomic predictors across both outcomes.

Together, these results underscore the multifactorial and biomedical nature of birth outcomes. The consistent importance of clinical indicators must be viewed alongside the structural and behavioral context in which pregnancies occur. Stratified findings by Hispanic subgroup suggest the need for more culturally nuanced interventions. For instance, while the “Hispanic paradox” has historically suggested protective health effects among immigrants, this concept may be outdated or overly simplistic. Our results reveal divergence across subgroups that likely reflect differences in nativity, acculturation, and migration history.

The extent to which these subgroups are first-, second-, or third-generation immigrants likely plays a role in health outcomes. Greater time in the U.S. may lead to physiological embodiment of structural disadvantage, as social stressors “get under the skin” and affect health. Additionally, trauma before or during migration and socioeconomic status (SES) likely interact with these outcomes. Unfortunately, our data lack measures on generation, immigration status, or trauma exposure—highlighting a major limitation. Moreover, Hispanic origin is based on self-report and subject to misclassification, which could bias subgroup estimates. These limitations should temper interpretations of subgroup differences and point to areas for future research.

Moving forward, studies should incorporate more nuanced measures of nativity, length of residence, acculturation, and neighborhood context to better understand the structural determinants of birth disparities. Integrating longitudinal data and qualitative insights could further elucidate how stress, social support, and health behaviors evolve over time to shape maternal and infant health.

## 5. Conclusions

This study shows that although Hispanic mothers as a group appear to have strong birth outcomes overall, there are important differences within the group—driven by medical conditions, personal behaviors, and broader social and structural factors. The findings call attention to how maternal risk unfolds along intersecting lines of ethnicity and generation. As public health shifts from a purely biomedical to a biosocial framework, our results underscore the need to consider how social determinants—such as education, insurance access, migration history, and community context—become biologically embedded across time and generations. Disaggregated analysis is not simply a methodological choice but a public health imperative, necessary to design responsive interventions and equitable maternal care systems. Future policies must recognize the heterogeneity within Hispanic communities, addressing not only proximal risk factors but also upstream conditions that shape reproductive health across lifespans and sociopolitical borders.

## Figures and Tables

**Figure 1 ijerph-22-01325-f001:**
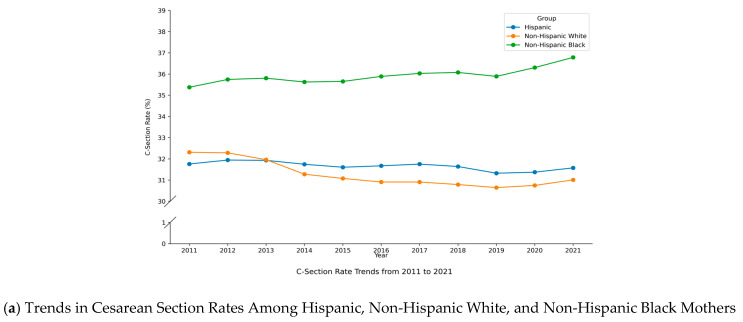
Trends in C-Section, Low Birthweight, and Prematurity Rates Among Hispanic, Non-Hispanic Black, and Non-Hispanic White Mothers (2011–2021). (**a**) C-section Rate; (**b**) Low Birthweight Rate; (**c**) Prematurity Rate. Data source: Authors’ calculations based on data from NCHS.

**Figure 2 ijerph-22-01325-f002:**
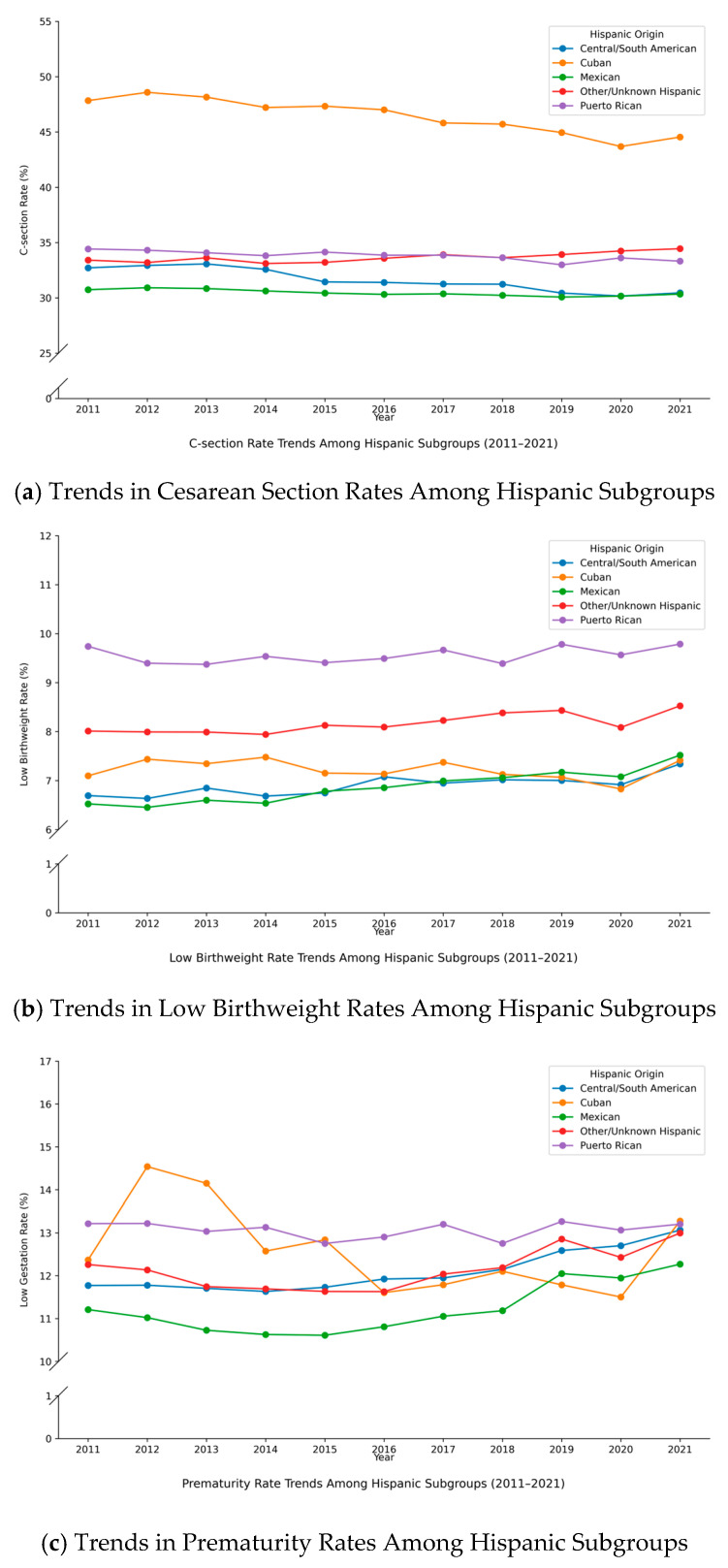
Trends in C-Section, Low Birthweight, and Prematurity Rates Among Different Hispanic Origin (2011–2021). (**a**) C-section Rate; (**b**) Low Birthweight Rate; (**c**) Prematurity Rate. Data source: Authors’ calculations based on data from NCHS.

**Figure 3 ijerph-22-01325-f003:**
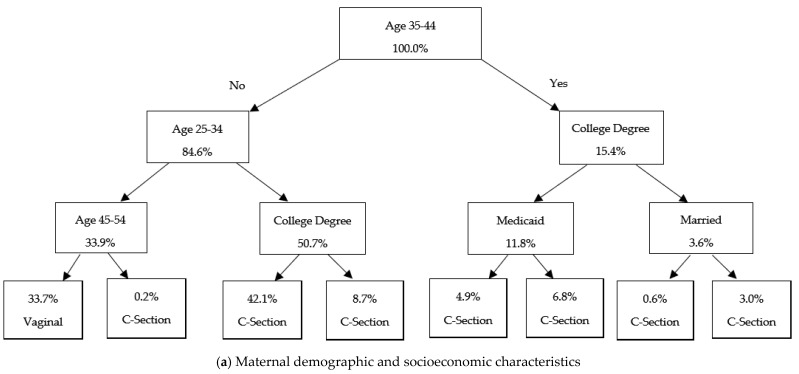
Classification Tree Models for Predicting C-Section Among Hispanics. Note: This figure presents classification tree models predicting C-section delivery, using different groups of control variables. The trees are constructed using the Gini impurity criterion to determine optimal splits. Each node represents a decision rule based on a control variable with values indicating the proportion of observations assigned to that category. In the terminal nodes, “Vaginal” and “C-section” indicate the model’s predicted mode of delivery.

**Figure 4 ijerph-22-01325-f004:**
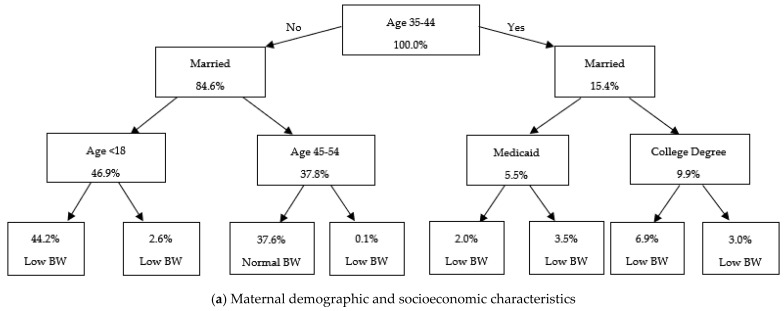
Classification tree for low birthweight among Hispanic births. Note: This figure presents classification tree models predicting low birthweight, using different groups of control variables. The trees are constructed using the Gini impurity criterion to determine optimal splits. Each node represents a decision rule based on a control variable, with values indicating the predicted outcome, and nodes, “Low BW” and “Normal BW”, indicate the model’s predicted birthweight category.

**Figure 5 ijerph-22-01325-f005:**
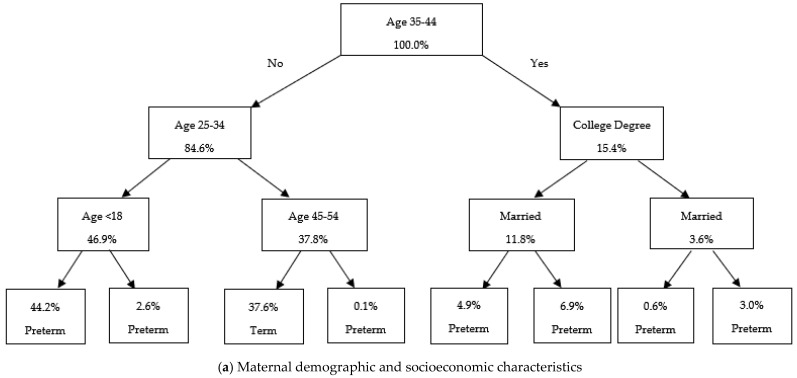
Classification tree for prematurity among Hispanic births. Note: This figure presents classification tree models predicting prematurity, based on different groups of control variables. The trees are constructed using the Gini impurity criterion to identify optimal splits. Each node represents a decision rule derived from a control variable, with values indicating the proportion of observations assigned to that category. In the terminal nodes, “Preterm” and “Term” indicate the model’s predicted gestational age category.

**Table 1 ijerph-22-01325-t001:** Summary Statistics by Hispanic Origins.

Panel A. Outcome Variables and Maternal Demographic and Socioeconomic Characteristics								
	C-Section Rate (%)	Low Birth Weight Rate (%)	Prematurity Rate (%)	Age (Years)	Bachelor’s Degree or Higher (%)		Married (%)	Medicaid Use (%)
All Hispanics	31.7	7.3	11.7	27.8	12.8		42.4	57.2
Mexican	30.5	6.9	11.2	27.5	10.1		42.6	58.5
Puerto Rican	33.8	9.6	13.1	27.0	15.2		35.4	55.7
Cuban	46.3	7.2	12.5	29.2	28.2		48.3	51.3
Central/South American	31.4	6.9	12.2	29.1	16.9		45.2	51.4
Other/Unknown	33.7	8.2	12.1	27.6	16.0		40.5	60.9
Panel B. Infant and birth characteristics								
	Gestational age (weeks)	Birthweight (grams)	Singleton (%)	First birth (%)	Breech (%)		Macrosomia (%)	Male (%)
All Hispanics	38.6	3269.8	97.5	28.9	1.9		7.0	51.0
Mexican	38.7	3284.8	97.7	28.3	1.7		7.3	51.0
Puerto Rican	38.5	3200.6	96.8	29.7	2.1		5.9	51.1
Cuban	38.5	3289.4	96.9	33.4	2.4		7.5	51.3
Central/South American	38.6	3269.5	97.6	28.6	2.1		6.7	50.9
Other/Unknown	38.5	3234.3	97.2	31.0	2.7		6.3	51.0
Panel C. Maternal health behavior and physical health characteristics								
	Obese (%)	Weight Gain (pounds)	Prenatal Care starts in first trimester (%)	Smoking during pregnancy (%)	Infertility treatment (%)			Previous C-section (%)
All Hispanics	27.8	30.3	67.3	1.6	0.6			16.1
Mexican	29.4	29.5	67.4	1.3	0.5			16.1
Puerto Rican	29.4	33.1	68.2	5.4	0.9			15.4
Cuban	20.5	34.3	77.7	2.0	1.0			17.7
Central/South American	20.7	30.3	62.1	0.5	0.8			15.4
Other/Unknown	30.2	31.2	71.8	2.1	0.8			17.5
Panel D. Maternal medical conditions and infections								
	Pre-pregnancy diabetes (%)	Gestational diabetes	Pre-pregnancy hypertension	Gestational hypertension	Hypertension/eclampsia	Gonorrhea	Hepatitis B	Hepatitis C
	(%)	(%)	(%)	(%)	(%)	(%)	(%)	(%)
All Hispanics	6.6	6.6	5.1	5.1	0.2	0.2	0.1	0.1
Mexican	6.9	6.9	4.9	4.9	0.2	0.2	0.0	0.1
Puerto Rican	6.2	6.2	5.7	5.7	0.3	0.4	0.2	0.4
Cuban	5.7	5.7	6.0	6.0	0.2	0.1	0.1	0.2
Central/South American	6.1	6.1	4.4	4.4	0.2	0.1	0.1	0.1
Other/Unknown	6.4	6.4	6.2	6.2	0.3	0.2	0.1	0.2

**Table 2 ijerph-22-01325-t002:** Logistic Regression Results for C-section.

	(1)	(2)	(3)	(4)	(5)	(6)
Control Variables	All Hispanics	Mexican	Puerto Rican	Cuban	Central South American	Other
Mothers Age	0.016 ***	0.012 ***	0.017 ***	0.039 ***	0.025 ***	0.017 ***
	(0.002)	(0.001)	(0.002)	(0.001)	(0.003)	(0.002)
Maternal College Degree	0.270 ***	0.275 ***	0.149 ***	−0.053	0.356 ***	0.125 ***
	(0.017)	(0.025)	(0.015)	(0.036)	(0.034)	(0.019)
Marital Status	−0.107 ***	−0.108 ***	−0.142 ***	−0.292 ***	−0.016	−0.107 ***
	(0.008)	(0.006)	(0.017)	(0.018)	(0.019)	(0.028)
Medicaid	−0.068 ***	−0.070 ***	−0.155 ***	0.004	−0.032	−0.128 ***
	(0.014)	(0.014)	(0.022)	(0.023)	(0.027)	(0.047)
Singleton	−2.165 ***	−2.254 ***	−1.886 ***	−2.218 ***	−2.137 ***	−2.137 ***
	(0.080)	(0.092)	(0.050)	(0.133)	(0.060)	(0.081)
Breech	3.713 ***	3.790 ***	3.505 ***	3.503 ***	3.804 ***	3.564 ***
	(0.203)	(0.263)	(0.068)	(0.076)	(0.119)	(0.184)
Infant Male	0.120 ***	0.118 ***	0.133 ***	0.128 ***	0.122 ***	0.121 ***
	(0.004)	(0.007)	(0.007)	(0.008)	(0.007)	(0.006)
Fetal Macrosomia	0.545 ***	0.532 ***	0.563 ***	0.681 ***	0.599 ***	0.581 ***
	(0.026)	(0.033)	(0.020)	(0.062)	(0.022)	(0.027)
Prenatal Care Began in 1st Trimester	0.099 ***	0.086 ***	0.075 ***	0.025	0.107 ***	0.076 **
	(0.019)	(0.025)	(0.013)	(0.021)	(0.018)	(0.031)
Pre-pregnancy Obesity	0.428 ***	0.454 ***	0.493 ***	0.352 ***	0.305 ***	0.425 ***
	(0.017)	(0.017)	(0.013)	(0.021)	(0.009)	(0.025)
Weight Gain	0.013 ***	0.013 ***	0.011 ***	0.010 ***	0.012 ***	0.012 ***
	(0.001)	(0.001)	(0.000)	(0.001)	(0.000)	(0.001)
Smoking during Pregnancy	0.168 ***	0.202 ***	0.061 ***	−0.071	0.214 ***	0.082 ***
	(0.021)	(0.030)	(0.019)	(0.043)	(0.049)	(0.017)
Infertility Treatment Used	0.789 ***	0.800 ***	0.739 ***	0.693 ***	0.849 ***	0.682 ***
	(0.021)	(0.013)	(0.025)	(0.049)	(0.039)	(0.046)
Pre-pregnancy Diabetes	0.741 ***	0.782 ***	0.856 ***	0.503 ***	0.598 ***	0.747 ***
	(0.023)	(0.026)	(0.044)	(0.037)	(0.041)	(0.046)
Gestational Diabetes	0.254 ***	0.267 ***	0.327 ***	0.265 ***	0.213 ***	0.249 ***
	(0.009)	(0.012)	(0.013)	(0.017)	(0.016)	(0.012)
Pre-pregnancy Hypertension	0.584 ***	0.604 ***	0.486 ***	0.499 ***	0.568 ***	0.549 ***
	(0.013)	(0.015)	(0.022)	(0.061)	(0.028)	(0.032)
Gestational Hypertension	0.656 ***	0.679 ***	0.515 ***	0.608 ***	0.653 ***	0.651 ***
	(0.027)	(0.025)	(0.024)	(0.027)	(0.019)	(0.050)
Hypertension Eclampsia	1.005 ***	1.042 ***	0.788 ***	0.921 ***	1.040 ***	1.039 ***
	(0.044)	(0.056)	(0.095)	(0.108)	(0.079)	(0.057)
Previous C-Section	3.508 ***	3.607 ***	3.219 ***	3.609 ***	3.296 ***	3.601 ***
	(0.144)	(0.173)	(0.056)	(0.148)	(0.078)	(0.125)
Gonorrhea	0.008	0.004	−0.050	0.059	0.060	−0.036
	(0.031)	(0.055)	(0.046)	(0.114)	(0.082)	(0.044)
Hepatitis B	−0.070	−0.113 **	−0.317 ***	0.244	−0.035	−0.050
	(0.065)	(0.049)	(0.106)	(0.240)	(0.075)	(0.108)
Hepatitis C	0.094 **	0.210 ***	0.110 *	−0.070	0.147 **	−0.072
	(0.048)	(0.037)	(0.066)	(0.115)	(0.071)	(0.074)
Observations	9,321,637	5,431,675	671,725	217,958	1,430,524	1,419,349

Note: Coefficients are reported as odds ratios. Robust standard errors are shown in parentheses. All models control for state fixed effects, year fixed effects, and the day of the week of birth. *** *p* < 0.01, ** *p* < 0.05, * *p* < 0.10.

**Table 3 ijerph-22-01325-t003:** Logistic Regression Results for Low Birth Weight.

	(1)	(2)	(3)	(4)	(5)	(6)
Control Variables	All Hispanics	Mexican	Puerto Rican	Cuban	Central South American	Other
Mothers Age	0.000	0.001	0.006 ***	0.005 *	−0.001	0.003 ***
	(0.001)	(0.001)	(0.001)	(0.002)	(0.001)	(0.001)
Maternal College Degree	−0.067 ***	−0.086 ***	−0.150 ***	−0.112 ***	−0.036 **	−0.107 ***
	(0.011)	(0.010)	(0.014)	(0.020)	(0.015)	(0.017)
Marital Status	−0.157 ***	−0.149 ***	−0.184 ***	−0.147 ***	−0.086 ***	−0.169 ***
	(0.011)	(0.017)	(0.017)	(0.015)	(0.012)	(0.014)
Medicaid	0.027 **	0.029 ***	0.050 ***	0.003	−0.005	0.003
	(0.012)	(0.010)	(0.019)	(0.032)	(0.026)	(0.023)
Singleton	−3.270 ***	−3.271 ***	−3.137 ***	−3.381 ***	−3.339 ***	−3.258 ***
	(0.015)	(0.019)	(0.030)	(0.042)	(0.024)	(0.021)
Breech	1.158 ***	1.176 ***	1.182 ***	1.003 ***	1.119 ***	1.161 ***
	(0.019)	(0.025)	(0.032)	(0.055)	(0.047)	(0.040)
Infant Male	−0.108 ***	−0.082 ***	−0.168 ***	−0.200 ***	−0.128 ***	−0.126 ***
	(0.013)	(0.012)	(0.011)	(0.013)	(0.014)	(0.019)
Prenatal Care Began in 1st Trimester	0.020 *	0.020 **	−0.046 **	0.003	0.059 ***	−0.027
	(0.011)	(0.009)	(0.021)	(0.049)	(0.022)	(0.018)
Pre-pregnancy Obesity	−0.424 ***	−0.429 ***	−0.549 ***	−0.404 ***	−0.340 ***	−0.457 ***
	(0.012)	(0.015)	(0.021)	(0.019)	(0.010)	(0.017)
Weight Gain	−0.033 ***	−0.033 ***	−0.033 ***	−0.035 ***	−0.034 ***	−0.033 ***
	(0.000)	(0.001)	(0.001)	(0.001)	(0.001)	(0.001)
Smoking during Pregnancy	0.839 ***	0.845 ***	0.598 ***	0.961 ***	1.023 ***	0.769 ***
	(0.057)	(0.073)	(0.030)	(0.127)	(0.109)	(0.054)
Infertility Treatment Used	0.334 ***	0.359 ***	0.354 ***	0.156 **	0.295 ***	0.304 ***
	(0.018)	(0.028)	(0.049)	(0.075)	(0.039)	(0.033)
Pre-pregnancy Diabetes	0.228 ***	0.224 ***	0.255 ***	0.254 ***	0.154 ***	0.264 ***
	(0.019)	(0.020)	(0.030)	(0.091)	(0.041)	(0.038)
Gestational Diabetes	−0.121 ***	−0.108 ***	−0.198 ***	−0.104 ***	−0.093 ***	−0.141 ***
	(0.010)	(0.011)	(0.019)	(0.027)	(0.019)	(0.012)
Pre-pregnancy Hypertension	1.321 ***	1.346 ***	1.074 ***	1.155 ***	1.442 ***	1.275 ***
	(0.018)	(0.021)	(0.019)	(0.048)	(0.044)	(0.033)
Gestational Hypertension	1.450 ***	1.501 ***	1.186 ***	1.406 ***	1.528 ***	1.358 ***
	(0.042)	(0.043)	(0.061)	(0.025)	(0.032)	(0.062)
Hypertension Eclampsia	1.764 ***	1.815 ***	1.496 ***	2.025 ***	1.819 ***	1.733 ***
	(0.068)	(0.081)	(0.129)	(0.118)	(0.072)	(0.090)
Previous C-Section	0.024 *	0.048 ***	−0.029	−0.228 ***	0.019	−0.007
	(0.014)	(0.009)	(0.033)	(0.030)	(0.021)	(0.022)
Gonorrhea	0.159 ***	0.129 ***	0.097	0.576 ***	0.144 **	0.166 ***
	(0.031)	(0.034)	(0.088)	(0.171)	(0.066)	(0.052)
Hepatitis B	0.016	−0.133	−0.016	−0.041	0.092	0.093
	(0.078)	(0.138)	(0.183)	(0.261)	(0.085)	(0.160)
Hepatitis C	0.434 ***	0.392 ***	0.390 ***	0.286	0.201 *	0.436 ***
	(0.039)	(0.061)	(0.037)	(0.196)	(0.121)	(0.079)
Observations	9,324,315	5,432,990	671,949	218,023	1,430,865	1,419,575

Note: Coefficients are reported as odds ratios. Robust standard errors are shown in parentheses. All models control for state fixed effects, year fixed effects, and the day of the week of birth. *** *p* < 0.01, ** *p* < 0.05, * *p* < 0.10.

**Table 4 ijerph-22-01325-t004:** Logistic Regression Results for Prematurity.

	(1)	(2)	(3)	(4)	(5)	(6)
Control Variables	All Hispanics	Mexican	Puerto Rican	Cuban	Central South American	Other
Mothers Age	0.0114 ***	0.0122 ***	0.0164 ***	0.0132 ***	0.00763 ***	0.0125 ***
	(0.00853)	(0.00106)	(0.000705)	(0.00108)	(0.00104)	(0.000824)
Maternal College Degree	−0.161 ***	−0.176 ***	−0.171 ***	−0.0425	−0.175 ***	−0.177 ***
	(0.0177)	(0.0126)	(0.0137)	(0.0414)	(0.0222)	(0.0130)
Marital Status	−0.162 ***	−0.163 ***	−0.148 ***	−0.116 ***	−0.144 ***	−0.140 ***
	(0.00406)	(0.00484)	(0.00909)	(0.0274)	(0.00770)	(0.00990)
Medicaid	0.0608 ***	0.0650 ***	0.0712 ***	0.0530 **	0.0187	0.0625 ***
	(0.00868)	(0.00718)	(0.0174)	(0.0249)	(0.0252)	(0.0216)
Singleton	−2.557 ***	−2.574 ***	−2.516 ***	−2.332 ***	−2.511 ***	−2.612 ***
	(0.0496)	(0.0553)	(0.0305)	(0.0659)	(0.0482)	(0.0334)
Breech	0.907 ***	0.918 ***	0.978 ***	0.646 ***	0.827 ***	0.950 ***
	(0.0333)	(0.0357)	(0.0463)	(0.0867)	(0.0423)	(0.0381)
Infant Male	0.152 ***	0.168 ***	0.117 ***	0.0892 **	0.141 ***	0.141 ***
	(0.0100)	(0.0125)	(0.00963)	(0.00792)	(0.0110)	(0.00909)
Prenatal Care Began in 1st Trimester	−0.148 ***	−0.142 ***	−0.166 ***	−0.0171	−0.195 ***	−0.154 ***
	(0.0244)	(0.0269)	(0.0337)	(0.0434)	(0.0273)	(0.0311)
Pre-pregnancy Obesity	−0.143 ***	−0.138 ***	−0.253 ***	−0.00846	−0.106 ***	−0.174 ***
	(0.0158)	(0.0222)	(0.0188)	(0.0295)	(0.0103)	(0.00941)
Weight Gain	−0.0172 ***	−0.0170 ***	−0.0184 ***	−0.0114 ***	−0.0177 ***	−0.0179 ***
	(0.000869)	(0.000978)	(0.000643)	(0.00223)	(0.000971)	(0.00104)
Smoking during Pregnancy	0.476 ***	0.520 ***	0.307 ***	0.506 ***	0.584 ***	0.435 ***
	(0.0532)	(0.0680)	(0.0204)	(0.0823)	(0.0840)	(0.0503)
Infertility Treatment Used	0.334 ***	0.346 ***	0.373 ***	0.187 **	0.305 ***	0.347 ***
	(0.0204)	(0.0249)	(0.0515)	(0.0765)	(0.0416)	(0.0302)
Pre-pregnancy Diabetes	0.640 ***	0.627 ***	0.804 ***	0.537 ***	0.542 ***	0.678 ***
	(0.0151)	(0.0167)	(0.0338)	(0.114)	(0.0324)	(0.0207)
Gestational Diabetes	0.162 ***	0.174 ***	0.146 ***	0.117 ***	0.147 ***	0.153 ***
	(0.0159)	(0.0188)	(0.0226)	(0.0233)	(0.0110)	(0.0232)
Pre-pregnancy Hypertension	0.913 ***	0.925 ***	0.820 ***	0.707 ***	0.936 ***	0.930 ***
	(0.0236)	(0.0288)	(0.0191)	(0.0833)	(0.0408)	(0.0406)
Gestational Hypertension	1.003 ***	1.045 ***	0.845 ***	0.899 ***	0.952 ***	1.000 ***
	(0.0469)	(0.0508)	(0.0498)	(0.0462)	(0.0292)	(0.0566)
Hypertension Eclampsia	1.346 ***	1.355 ***	1.271 ***	1.314 ***	1.357 ***	1.362 ***
	(0.0316)	(0.0361)	(0.0707)	(0.123)	(0.0376)	(0.0754)
Previous C-Section	0.198 ***	0.212 ***	0.152 ***	0.0682 *	0.194 ***	0.185 ***
	(0.0107)	(0.00701)	(0.0217)	(0.0368)	(0.0137)	(0.0153)
Gonorrhea	0.156 ***	0.153 ***	0.135 **	0.163	0.152 *	0.135 **
	(0.0402)	(0.0426)	(0.0620)	(0.132)	(0.0828)	(0.0674)
Hepatitis B	−0.00760	0.0286	−0.0646	−0.428 **	0.0143	−0.0475
	(0.0678)	(0.0672)	(0.169)	(0.215)	(0.0748)	(0.112)
Hepatitis C	0.316 ***	0.390 ***	0.269 ***	0.308 **	0.248 ***	0.226 ***
	(0.0331)	(0.0368)	(0.0397)	(0.135)	(0.0820)	(0.0737)
Observations	9,324,315	5,432,990	671,949	218,023	1,430,865	1,419,575

Note: Coefficients are reported as odds ratios. Robust standard errors are shown in parentheses. All models control for state fixed effects, year fixed effects, and the day of the week of birth. *** *p* < 0.01, ** *p* < 0.05, * *p* < 0.10.

## Data Availability

The data used in this study are derived from the National Vital Statistics System (NVSS) Natality public-use and restricted-use datasets, provided by the National Center for Health Statistics (NCHS). The public-use data are available at https://www.cdc.gov/nchs/data_access/vitalstatsonline.htm. Access to restricted-use files requires an approved data use agreement through the NCHS Research Data Center due to confidentiality protections. No new data were created for this study. The data is accessed on 1 May 2024.

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
