# Peer review of "Birth Outcomes in the Hispanic Population in the United States: Trends, Variation, and Determinants (2011–2021)"

_ijerph, 2025, doi:10.3390/ijerph22091325_

Round 1

Reviewer 1 Report

Comments and Suggestions for Authors

Explain why the absence of the analysis described below is secondary to what you did. Or explain why your study is novel, helpful, moves us forward or adds to what we are trying to do in public health.   It seems you only manipulated data with no theories on why the data are as you have demonstrated.  The control variables are not novel only using medical and healthcare factors that don't go far enough upstream to determine structural systemic interventions.  In this study, we build upon previous research by documenting the rates of C-section, low birthweight, and prematurity .  In public health we are way past this kind of epidemiology and onto a biosocial model of health that looks upstream at causal patterns related to the non-medical social and environmental issues and SDOH.   Develop in discussion theories such as these as they are in the literature: There are great differences in Hispanics in the U.S. as the authors point out.    Are they 1st 2nd or 3rd generation? Any ideas on how that make a difference?  From where did they originate and how long have they been in the US?  Did they come here experiencing trauma from their homeland only to find more here in the US? What is their SES status?  Are they low middle or upper income?  Already mentioned Hispanic paradox but that theory is 35 years old has it been developed?  The longer they are in the US the more they become American in the way they think, behave and ultimately due to movement of the external across the CNS into the body where these external factors get under the skin to change them physiologically.  see epigenetics, Geronimus, and Nancy Krieger, 

Author Response

Comment 1: Explain why the absence of the analysis described below is secondary to what you did (see epigenetics, Geronimus, and Nancy Krieger). Or explain why your study is novel, helpful, moves us forward or adds to what we are trying to do in public health.
Response 1:
Thank you for pointing this out. Our research advances current knowledge by examining trends over time and disaggregating data by Hispanic origin subgroups using a nationally representative dataset. Prior studies often lack this level of granularity or rely on localized data (e.g., single-state, hospital-based samples), which may limit generalizability and introduce potential biases. In contrast, our analysis uses data covering nearly all births in the U.S., offering broader insight into population-level patterns.

We acknowledge that our dataset is limited in key areas—it does not include genetic data, immigration history, or detailed information on lifetime health behaviors. These are indeed critical to understanding the biosocial context and cumulative disadvantage described in work by Geronimus and Krieger. We have incorporated discussion of potential immigration-related influences into the revised manuscript and appreciate the encouragement to reflect on these dimensions.

Comment 2: It seems you only manipulated data with no theories on why the data are as you have demonstrated. The control variables are not novel only using medical and healthcare factors that don't go far enough upstream to determine structural systemic interventions. In this study, we build upon previous research by documenting the rates of C-section, low birthweight, and prematurity. In public health we are way past this kind of epidemiology and onto a biosocial model of health that looks upstream at causal patterns related to the non-medical social and environmental issues and SDOH.
Response 2:
We appreciate this important critique. While our study primarily uses clinical and behavioral variables available in vital records, our intent is not to overlook upstream determinants but to establish a population-level baseline for subgroup differences within the Hispanic population. The absence of structural and biosocial variables in our dataset is a limitation we now emphasize more explicitly in the discussion.

That said, our findings lay the groundwork for future biosocial research by identifying which subgroups are at heightened risk for adverse birth outcomes. We now also point readers to the relevance of structural racism, immigration status, acculturation, and other social determinants of health (SDOH) that may contribute to these disparities, even if they cannot be measured directly in our data. These considerations align with the biosocial health model, and we have added theoretical framing in the revised discussion to reflect this.

Comment 3: Develop in discussion theories such as these as they are in the literature: There are great differences in Hispanics in the U.S. as the authors point out. Are they 1st 2nd or 3rd generation? Any ideas on how that make a difference? From where did they originate and how long have they been in the US? Did they come here experiencing trauma from their homeland only to find more here in the US? What is their SES status? Are they low middle or upper income? Already mentioned Hispanic paradox but that theory is 35 years old has it been developed? The longer they are in the US the more they become American in the way they think, behave and ultimately due to movement of the external across the CNS into the body where these external factors get under the skin to change them physiologically.
Response 3:
Thank you for these thought-provoking comments. We agree that understanding generational status, immigration experience, trauma exposure, and socioeconomic context is essential to interpreting variation in birth outcomes. Unfortunately, the NVSS data lack indicators such as generation, years in the U.S., or migration-related trauma. However, we have revised the discussion to include these theoretical frameworks and hypotheses.

We expand on how acculturation, stress biology, and embodied structural disadvantage may contribute to differences across subgroups, including the potential evolution of the Hispanic paradox over time. We also acknowledge that assimilation can lead to both protective and harmful changes in behavior and physiology, as described in work on "weathering" and embodiment. These additions strengthen the theoretical grounding of our analysis and address the deeper public health implications raised in your comments.

Reviewer 2 Report

Comments and Suggestions for Authors
  1. Line 11: Add “d” to the word, “associate.” 
  2. Line 21: Change “Puerto Rican mothers show higher rate of low birthweight and prematurity rate” to “Puerto Rican mothers show higher rates of low birthweight and prematurity.”
  3. Line 21: When first mentioning “Cesarean section,” in the abstract, provide the acronym (C-section) in parentheses.
  4. Classification Tree Models: In the abstract, consider adding one sentence to address which maternal, infant, or clinical factors were most predictive in the classification tree model. Address whether the tree revealed results beyond what was observed in the logistic regression.
  5. Conclusion: Consider revising the conclusion to avoid repeating content from the results and discussion. Ideally, use the conclusion to clearly highlight the study's broader implications and potential applications.
  6. Limitations: Include 1-2 sentences on how limitations eg  potential misclassification of Hispanic origin could have impacted study findings and how readers interpret the findings? 
  7. General: There are grammatical errors throughout the paper. Careful proofreading is recommended. Additionally, the authors use a mix of past and present tense throughout the manuscript. Consider revising for consistency, ideally using past tense when referring to study methods and results.

Author Response

Comment 1: Line 11: Add “d” to the word, “associate.”
Response 1: Thank you for pointing this out. The correction has been made.

Comment 2: Line 21: Change “Puerto Rican mothers show higher rate of low birthweight and prematurity rate” to “Puerto Rican mothers show higher rates of low birthweight and prematurity.”
Response 2: Thank you. We have revised the sentence accordingly for grammatical accuracy and clarity.

Comment 3: Line 21: When first mentioning “Cesarean section,” in the abstract, provide the acronym (C-section) in parentheses.
Response 3: Thanks for the suggestion. We updated Line 10 in the abstract to introduce the acronym upon first mention: "Cesarean section (C-section)."

Comment 4: Classification Tree Models: In the abstract, consider adding one sentence to address which maternal, infant, or clinical factors were most predictive in the classification tree model. Address whether the tree revealed results beyond what was observed in the logistic regression.
Response 4: We appreciate this helpful suggestion. We have added the following sentence to the abstract:
“Logistic regression results highlight multiple births, breech presentation, and hypertensive conditions as key factors associated with adverse birth outcomes. Complementing these findings, our classification tree analysis identifies inadequate gestational weight gain (less than 15 pounds) as a prominent risk factor for both low birthweight and prematurity. Additionally, obesity emerges as a significant factor linked to increased likelihood of C-section.”

Comment 5: Conclusion: Consider revising the conclusion to avoid repeating content from the results and discussion. Ideally, use the conclusion to clearly highlight the study's broader implications and potential applications.
Response 5: Thank you for the recommendation. We have revised the conclusion to emphasize the broader implications and potential applications of our findings rather than restating the results. The new conclusion is included in the revised manuscript.

Comment 6: Limitations: Include 1-2 sentences on how limitations e.g., potential misclassification of Hispanic origin could have impacted study findings and how readers interpret the findings.
Response 6: We have revised the limitations section to include the following:
“Potential misclassification of Hispanic origin may have introduced measurement error, which could attenuate observed differences across subgroups. This limitation should be considered when interpreting subgroup-specific findings.”

Comment 7: General: There are grammatical errors throughout the paper. Careful proofreading is recommended. Additionally, the authors use a mix of past and present tense throughout the manuscript. Consider revising for consistency, ideally using past tense when referring to study methods and results.
Response 7: Thank you for noting this. We have carefully proofread the manuscript and corrected grammatical issues. We have also revised the tense throughout for consistency, using past tense when describing methods and results.

Round 2

Reviewer 1 Report

Comments and Suggestions for Authors

At line 466 (see below) please change that word biosocial  to be biomedical as your study is biomedical . As you say later (tank you for this) we do need more biosocial analysis to greater contextual up stream determinants.  

In the abstract (add two sentences)  and introduction (4-5 sentences) please add summarizing text identifying the contrast in your study as a  biomedical approach versus a biosocial analysis.  

Together, these results underscore the multifactorial and biosocial nature of birth Line 466 
outcomes. The consistent importance of clinical indicators must be viewed alongside the 

Author Response

Reviewer Comment:
At line 466 (see below) please change that word biosocial to be biomedical as your study is biomedical. As you say later (thank you for this) we do need more biosocial analysis to greater contextual upstream determinants.

In the abstract (add two sentences) and introduction (4–5 sentences) please add summarizing text identifying the contrast in your study as a biomedical approach versus a biosocial analysis.

Author Response:
Thank you for this helpful clarification. We have revised the text at line 466 to replace biosocial with biomedical to accurately reflect the nature of our study. We agree that our analysis is primarily biomedical in approach, while acknowledging the importance of future biosocial research to explore upstream determinants of health.

As suggested, we have added two sentences to the abstract explicitly summarizing the contrast between our biomedical approach and a biosocial analysis, noting that our study focuses on physiological and clinical risk factors while recognizing the value of contextual social determinants. We have also incorporated 4–5 sentences in the introduction (last paragraph) to highlight this contrast in greater detail, framing our work as a biomedical analysis that integrates consideration of socioeconomic and demographic variables, and positioning biosocial research as a necessary complementary direction for future studies.